# Deep brain stimulation of subthalamic nucleus modulates cortical auditory processing in advanced Parkinson's Disease

Kati Valkonen[1,2], Jyrki P. Mäkelä[2], Katja Airaksinen[1], Jussi Nurminen[2], Riku Kivisaari[3], Hanna Renvall[2,4]*, Eero Pekkonen[1]

1 Department of Neurology, Helsinki University Hospital, Finland and Department of Clinical Neurosciences (Neurology), University of Helsinki, Helsinki, Finland, 2 BioMag Laboratory, Helsinki University Hospital Medical Imaging Center, Helsinki University Hospital, Helsinki University and Aalto University School of Science, Helsinki, Finland, 3 Department of Neurosurgery, Helsinki University Hospital, Helsinki, Finland, 4 Department of Neuroscience and Biomedical Engineering, Aalto University, Espoo, Finland

* hanna.renvall@hus.fi

**Data Availability Statement:** The original datasets are not publicly available due to restrictions placed by the Helsinki University Hospital Research Ethics Committee, as the data contains potentially

## Abstract

Deep brain stimulation (DBS) has proven its clinical efficacy in Parkinson's disease (PD), but its exact mechanisms and cortical effects continue to be unclear. Subthalamic (STN) DBS acutely modifies auditory evoked responses, but its long-term effect on auditory cortical processing remains ambiguous. We studied with magnetoencephalography the effect of long-term STN DBS on auditory processing in patients with advanced PD. DBS resulted in significantly increased contra-ipsilateral auditory response latency difference at ~100 ms after stimulus onset compared with preoperative state. The effect is likely due to normalization of neuronal asynchrony in the auditory pathways. The present results indicate that STN DBS in advanced PD patients has long-lasting effects on cortical areas outside those confined to motor processing. Whole-head magnetoencephalography provides a feasible tool to study motor and non-motor neural networks in PD, and to track possible changes related to cortical reorganization or plasticity induced by DBS.

## Introduction

Parkinson's disease (PD) is a progressive extrapyramidal movement disorder with main motor symptoms of rigidity, hypokinesia, and resting tremor. PD patients often suffer from a broad spectrum of non-motor signs, which may precede appearance of motor symptoms [1]. Degeneration of dopaminergic neurons in the substantia nigra is known to be mainly responsible for the parkinsonian symptoms, but the disease affects also cholinergic, serotonergic, and noradrenergic neurotransmission in cortical areas outside the motor system [2], likely related to the wide variety of patients' symptom profiles.

Despite optimal oral drug treatment, about 90% of PD patients develop severe motor fluctuations and/or dyskinesia within 5–10 years from the diagnosis [3]. In these patients, deep brain stimulation (DBS) of the subthalamic nucleus (STN) has been demonstrated to be an

identifying patient information. The data that support the findings of this study are available with permission of the Helsinki University Hospital Research Ethics Committee (eettiset. toimikunnat@hus.fi).

**Funding:** KV: Finnish Parkinson Foundation KV: University of Helsinki HR: grant 321460, Academy of Finland The funders had no role in study design, data collection and analysis, decision to publish, or preparation of the manuscript.

**Competing interests:** The authors have declared that no competing interests exist.

effective treatment [4–6]. However, the exact mechanisms of DBS have remained unclear. So far, there is no consensus whether the effect of DBS is local or system-wide, or whether DBS elicits mainly inhibition or excitation of the target nuclei [7–9]. It has been even hypothesized that DBS could be neuroprotective and slow down the degeneration of dopaminergic neurons in PD [10]. Recent observations from animal models suggest that the efficacy of DBS is likely to be mediated by multifactorial mechanisms, including immediate neuromodulatory effects such as inhibition of the neural soma and excitation of axons, and long-term effects such as neuronal reorganization, and synaptic plasticity [11]. DBS does not, however, arrest or reverse PD progression [12]. DBS also exerts effects on non-motor systems. For example, sleep was improved and anxiety alleviated by DBS during a 4-year follow-up in PD [13]. So far, no evidence on neuronal plasticity induced by DBS in humans is available.

Magnetoencephalography (MEG) provides a non-invasive and patient-friendly neuroimaging method for addressing possible cortical plasticity induced by DBS. The stimulator, however, causes strong artifacts in MEG recordings. With current data analysis methods, magnetic interference originating from sources close to the MEG sensors can be effectively suppressed [14, 15]; such methods have earlier been successfully used to remove magnetic artifacts caused by DBS [16–21] and vagus nerve stimulation (*e.g.*, [22]). This approach permits, with high temporal and spatial resolution, noninvasive measurements of cortical activity and of its possible modulations induced by DBS both within and outside the motor system. Indeed, MEG responses to speech sounds were recently demonstrated to be modified by DBS [23].

Alterations in auditory processing have been especially frequently described in PD patients (for a recent review, see [24]). Patients with PD have been demonstrated to suffer from impaired hearing compared with age-matched healthy control subjects (*e.g.*, [25]), and a previous MEG study suggested changes in the cortical processing of auditory information in PD patients without DBS [26]. DBS was subsequently shown to enhance the strength of the most prominent auditory evoked fields (AEFs) at ~100 ms after stimulus onset (N100m) [18], but the possible long-term modulation of auditory cortical responses by DBS has not been reported before.

Here we studied whether STN DBS has long lasting effects on auditory cortical processing in patients with advanced PD. We explored cortical activity elicited by simple tone stimuli, which produce well-characterized AEFs [27, 28]. We hypothesized that the long-term STN stimulation would enhance cortical auditory processing. Specifically, we anticipated that the STN stimulation would modify parallel cortical auditory processing between the hemispheres, in line with results on patients with unilateral conductive hearing loss demonstrating modified AEFs after middle ear surgery [29]. Particularly the latency differences between the contra- and ipsilateral auditory responses, *i.e.*, the interhemispheric asynchrony displays reorganization along with improved hearing [30]. Moreover, as AEF amplitudes display a large interindividual variability [31, 32], and they are sensitive to head movements in a long-term follow-up even when movement compensation is applied [33], we focused our analysis in DBS patients on the interhemispheric latency differences.

Our results show that the STN stimulation modifies parallel cortical auditory processing between the hemispheres, in line with the earlier results on patients with unilateral conductive hearing loss [29, 30].

## Materials and methods

Twenty-two (22) advanced PD patients who were screened for DBS implantation (including head MRI, levodopa challenge test, and thorough neuropsychological testing) originally participated in the study. None of the patients had dementia or severe depression. Clinical details of

the patients are shown in Table 1. The study was approved by the Ethics Committee of Helsinki University Central Hospital and all patients gave informed written consent prior the study.

The data of seven patients were rejected from further analysis. Three of these patients did not want to participate in the follow-up measurement at six months. One patient had a subdural haematoma due to a fall. The DBS device was removed from one patient due to an infection of the subcutaneous internal pulse generator. In the other two patients (one with a temporal bone titanium panel and another with exceptionally strong DBS artifacts) MEG amplifiers were saturated during the measurements, excluding the use of the MEG data. Three of the remaining 15 patients did not tolerate MEG measurement at six months when DBS was off, but their data during DBS on were included in the analyses.

The mean age of the remaining 15 patients (four females) was 55 years (range 36–67 years); see Table 1. They had received a diagnosis of PD on average 13 years (range 6–24 years) before the implantation of the bilateral STN DBS (Activa PC®, Medtronic, Minneapolis, Minnesota, United States). The MEG measurements were conducted 0.5–13 months (mean 7 months) before the DBS implantation and at 5–11 months (mean 7 months) after it. During the MEG recordings, the patients had their normal medication on (see Table 1). The mean Hoehn and Yahr scores [34] were 2.5 (range 2–3) both at the baseline and at seven months when medication was off and DBS on, suggesting bilateral nature of the disease, without or with only mild impairment of balance. The DBS frequency was adjusted to 130 Hz before MEG measurements to avoid interference with the head position indicator (HPI) coil signals. All patients were kept at their original stimulation settings during the MEG recordings. Monopolar DBS

**Table 1. Clinical details of patients and DBS parameters.**

| Patient | Sex | Age | PD duration before operation (yrs) | MEG measurements | | UPDRSIII | | LEDD (mg)* | | DBS | | | Freq (Hz) | Pulse width right/left (µs) |
|---|---|---|---|---|---|---|---|---|---|---|---|---|---|---|
| | | | | Time before DBS operation (mnths) | Time after DBS operation (mnths) | Before DBS | After DBS | Before DBS | After DBS | Voltage, right/left (V) | Bi- or monopolar, right/left | | | |
| 1 | F | 63 | 24 | 5 | 7 | 32 | 35 | 90 | 760 | 3.6/1.6 | bi/bi | 160 | 60/60 |
| 2 | M | 57 | 15 | 11 | 6 | 32 | 21 | 1 618 | 1381 | 2.5/2.5 | mono/bi | 130 | 60/60 |
| 3 | M | 63 | 8 | 3 | 5 | 37 | 29 | 925 | 639 | 2.5/2.5 | mono/mono | 160 | 60/60 |
| 4 | F | 56 | 18 | 0,5 | 7 | 74 | 34 | 1562 | 1386 | 2.6/2.7 | mono/mono | 130 | 60/60 |
| 5 | M | 62 | 17 | 8 | 5 | 37 | 33 | 1408 | 1407 | 3.6/3.1 | mono/mono | 130 | 60/60 |
| 6 | M | 67 | 9 | 5 | 7 | 46 | 25 | 1679 | 480 | 2.6/2.9 | mono/mono | 150 | 120/60 |
| 7 | F | 66 | 16 | 11 | 6 | 31 | 38 | 1292 | 1000 | 2.5/3.6 | mono/bi | 130 | 60/60 |
| 8 | M | 36 | 7 | 1 | 5 | 68 | 43 | 1 574 | 210 | 3.2/3.2 | mono/mono | 130 | 60/60 |
| 9 | M | 42 | 10 | 3 | 6 | 62 | 32 | 765 | 1497 | 3.2/3.2 | mono/mono | 130 | 60/60 |
| 10 | M | 45 | 9 | 5 | 5 | 31 | 29 | 1 481 | 1255 | 3.5/3.8 | bi/mono | 130 | 60/60 |
| 11 | F | 63 | 13 | 9 | 7 | 23 | 16 | 658 | 366 | 2.8/2.9 | mono/mono | 130 | 60/60 |
| 12 | M | 49 | 14 | 8 | 11 | 51 | 24 | 1 263 | 1164 | 3.5/3.1 | bi/mono | 150 | 60/60 |
| 13 | M | 47 | 8 | 3 | 6 | 37 | 24 | 655 | 580 | 2.3/2.5 | mono/mono | 180 | 60/60 |
| 14 | M | 42 | 6 | 5 | 6 | 44 | 20 | 1338 | 1384 | 3.5/2.0 | mono/mono | 130 | 60/60 |
| 15 | M | 62 | 18 | 4 | 7 | 27 | 20 | 1158 | 560 | 2.9/3.0 | mono/mono | 130 | 60/60 |
| **MEAN** | | **55** | **13** | **7** | **7** | **42** | **28** | **1222** | **938** | **3.0/2.8** | **-** | **140** | **64/60** |

DBS, deep brain stimulation; UPDRS, Unified Parkinson´s Rating Scale; Freq, stimulation frequency

* To calculate the levodopa equivalent daily dose (LEDD), the following formula was used:

100 mg l-dopa = 130 mg contolled-release l-dopa = 70 mg l-dopa + COMT inhibitor = 1 mg pramipexole = 5 mg ropinirole = 4 mg rotigotine.

induces more high-frequency artifacts than bipolar one, but they can be effectively removed by filtering (see, *e.g.*,[18]). Three in-hospital programming sessions with medication off preceded the DBS/MEG measurement for verifying the optimal DBS response.

The measurements were performed with the 306-channel Elekta Neuromag Vectorview® MEG device (Elekta Oy, Helsinki, Finland) in a magnetically shielded room (Euroshield, Eura, Finland). The baseline MEG was measured before the DBS implantation, and the follow-up measurements were conducted at about seven months (see above) after the operation, both with DBS on and off. During the MEG measurement, an experienced nurse accompanied the patient in the magnetically shielded room. Auditory stimulation consisted of 1-kHz sinusoidal 50-ms tone pips delivered through plastic tubes to each ear separately. Stimulus intensity was adjusted to be at a comfortable hearing level of > 60 dB HL, and all the patients reported having heard the pips as equally strong on both ears. The auditory stimulation was implemented as a part of a multimodal stimulation sequence that included also somatosensory and visual stimuli. The different stimulus types were presented in a pseudorandom order so that the same stimulus was not allowed to occur more than twice in a row to exclude formation of possible sensory memory traces. This resulted in a mean interstimulus interval (ISI) of 5.5 s for each stimulus type.

The recording passband was 0.03–330 Hz with a sampling rate of 1011 Hz. A vertical electro-oculogram (EOG) was recorded simultaneously for extracting eye-movement artifacts. The location of the head was determined by four indicator coils placed on the scalp; the exact head position with respect to the MEG sensor array was determined by briefly feeding current to the marker coils before the actual measurement. The location of the coils with respect to head landmarks was determined with a 3-D digitizer (Fastrak®, Polhemus, Inc., Colchester, Vermont, United States).

The strong magnetic artifacts caused by DBS were suppressed by the spatiotemporal signal space separation method (tSSS; [14]) using an 8-s time window and a subspace correlation limit of 0.9 [35]. The effect of tSSS on AEFs has been visualized and discussed in [18]. 111 ± 18 (mean ± SD) artifact-free auditory responses were averaged per stimulated ear. The responses were averaged from 100 ms before the stimulus onset to 500 ms after it, setting as baseline the 100-ms interval immediately preceding the stimulus onset, and filtered off-line at 1–40 Hz.

The 100-ms AEFs (N100m) we first analyzed at the sensor level. The peak response amplitudes were determined by finding the absolute maxima of evoked signals in a time window 80–130 ms after the stimulus onset at the gradiometer channels. The response latencies and amplitudes were then measured from the vector sum $\sqrt{\left(\frac{\partial Bz}{\partial x}\right)^2 + \left(\frac{\partial Bz}{\partial y}\right)^2}$ of the gradiometer pair showing the maximum signal. In signal strength comparisons, the vector sums simplify the analysis when the orientation of the neural current changes as a function of time, with only minor accompanying changes in the source location [36; for similar approach, see [37]]. In such a case, the amplitude measurements from any single channel can be misleading.

Subsequently, the cortical sources of the N100m responses were searched separately for contra- and ipsilateral hemispheres using a subset of 10–15 gradiometer pairs in both hemispheres, to adequately cover the loci of the response maxima by means of guided current modeling (equivalent current dipole [ECD]; [36]), separately for each subject. The model parameters were optimized for the intracranial space based on individual MR images that were available for all subjects. The N100m sources were estimated by a sequential ECD fitting using a 1-ms interval within the time period of 80–130 ms after the stimulus onset, separately for the data measured before DBS implantation and with DBS both on and off.

A two-dipole model (one in each hemisphere) was used to investigate the effect of DBS on AEFs in all three conditions (preoperational, DBS on, DBS off). The ECD corresponding to

the strongest source (and thus with best signal-to-noise ratio) was chosen from the three conditions (preoperational, DBS on, DBS off) to represent the source in all conditions, taken that the following requirements were fulfilled: 1) The dipole location was stable during 10 ms around the maximum ECD so that the variation of x-, y- and z-coordinates was less than 5 mm in each direction, 2) the dipole explained over 80% of the measured data variance (goodness of fit; g) of the selected channels, and 3) the maximum of the ECD amplitude peaked within the time period defined previously at the channel level. If two or more dipoles with same strengths fulfilled the criteria, the one with the best g-value was chosen. Applying the same source model in each data set minimizes variation due to possible differences between source models; we assumed that the locations of cortical representations would not be changed by DBS.

Statistical comparisons of the latency differences between the preoperative AEFs and those obtained with DBS on and DBS off were performed using non-parametric sign test which does not assume any particular value distribution. The results were Bonferroni corrected for number of comparisons (preoperational *vs.* DBS on, and preoperational *vs.* DBS off). The results are reported as mean ± standard error of mean.

## Results

Fig 1 demonstrates the AEFs at the sensor level in one patient in all three conditions (preoperative, DBS on, DBS off). After artifact removal by tSSS, sources of AEFs were analyzable in both hemispheres in all 15 patients, and well explained with two dipoles located bilaterally in the supratemporal cortices.

Table 2 summarizes the N100m response latencies and amplitudes (mean ± SEM) at the source level in all conditions (preoperational, DBS on, DBS off) for both stimulated ears. The ipsi-contralateral difference of N100m peak latencies significantly increased from the preoperative to DBS on condition (pooled across the stimulated ears, 10 ± 2 ms *vs.* 14 ± 1 ms: p = 0.036; see Fig 2). The ipsi-contralateral difference of N100m peak latencies did not increase statistically significantly in the DBS off condition compared to the preoperative state (pooled across the stimulated ears, 11 ± 1 ms *vs.* 13 ± 3 ms, p = 0.14).

Motor symptoms were effectively relieved by DBS when off medication. Mean motor Unified Parkinson's Disease Rating Scale part III (UPDRS-III) scores were 42 ± 15 before operation (medication off) and 28 ± 8 after DBS implantation (DBS on and medication off) at six months (n = 15; p = 0.005). The mean levodopa equivalent daily dose (LEDD) appeared to be decreased from 1222 ± 352 mg before operation to 938 ± 443 mg after DBS implantation (n = 15; p = 0.06).

## Discussion

This is the first follow-up MEG study with a relatively large number of advanced PD patients with STN DBS. Our results indicate that DBS induces long-term changes in auditory cortical processing, shown here by the significant increase in the ipsi-contralateral N100m peak latency difference for monaural stimulation, suggesting cortical reorganization related to the treatment with STN DBS.

Neural pathways from each ear project bilaterally, but dominantly to the contralateral auditory cortex. In healthy subjects, N100m responses are usually larger and peak earlier for contralateral than ipsilateral stimuli [38–40]. Signs of auditory cortical reorganization have earlier been observed after unilateral hearing loss: Both patients with congenital conductive hearing loss and with idiopathic sudden sensorineural hearing loss at adult age have earlier and stronger AEFs in the hemisphere ipsilateral to the stimulated healthy ear [28]. Similarly decreased

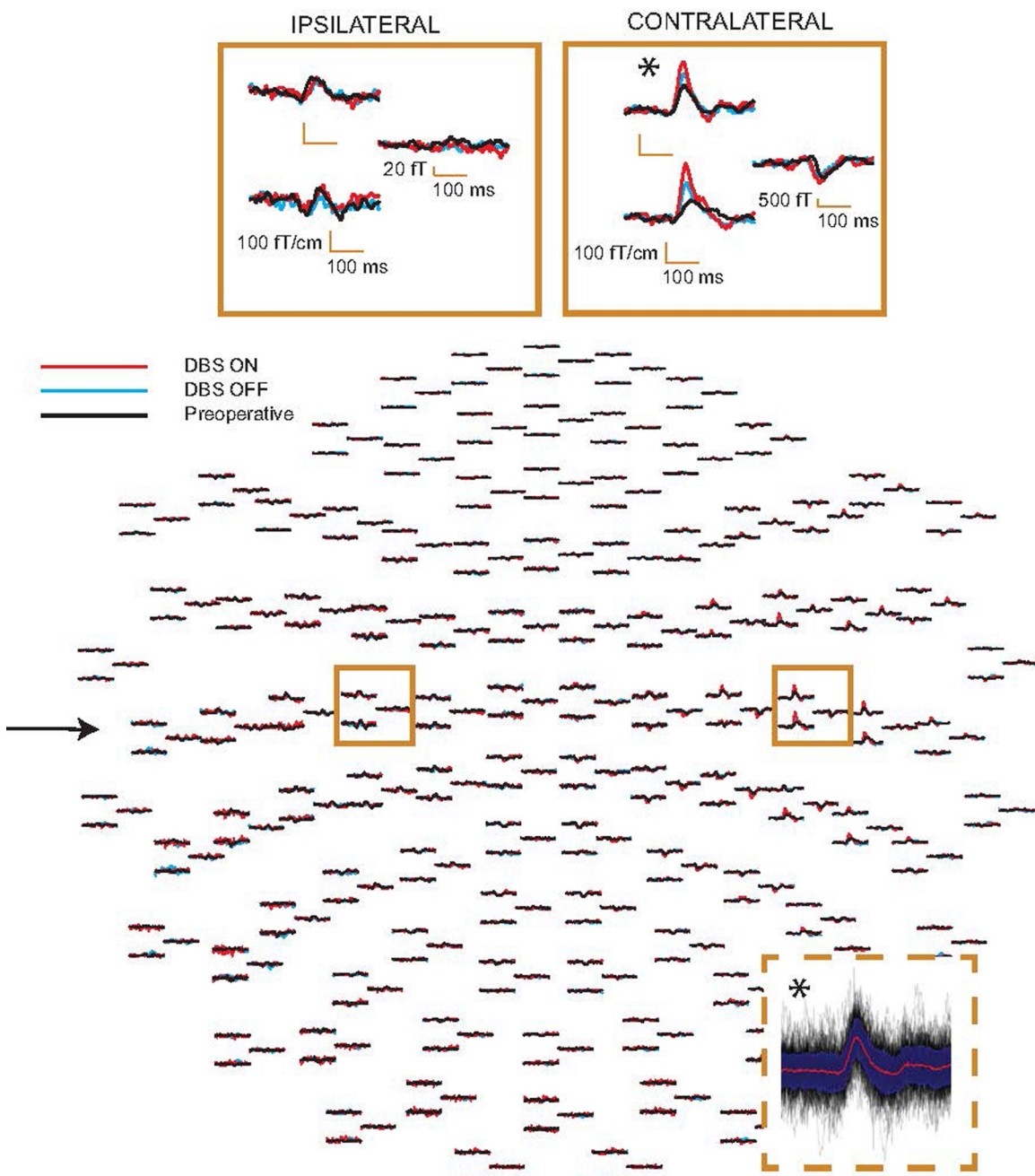

**Fig 1. Sensor level data in one patient.** Auditory responses to left-ear stimulation measured before DBS implantation (preoperative) and after the implantation, both DBS on and off. The arrow indicates the stimulated (left) ear, and the inserts (above) depict the maximum channels in the contra- and ipsilateral hemispheres. At each sensor triplet, the two left-sided sensors are gradiometers, and the right-sided one is a magnetometer. The insert (below) demonstrates the single-trial responses (black), their mean (red) and ± 1 SD (dark blue) in the preoperative condition at the maximum gradiometer channel (marked with asterisk).

contralateral dominance for unilateral stimulation has been observed in AEP [41] and in functional MRI studies [42] of unilaterally deaf subjects, and in patients with sudden hearing loss [43, 44]. On the other hand, AEFs were modified after middle ear surgery performed to correct unilateral conductive hearing loss: N100m peaked significantly earlier in the hemisphere

**Table 2. N100m response latencies and amplitudes.**

| Condition | N100m latencies (ms): Left-ear stimulation | | N100m latencies (ms): Right-ear stimulation | | N100m amplitudes (nAm): Left-ear stimulation | | N100m amplitudes (nAm): Right-ear stimulation | |
|---|---|---|---|---|---|---|---|---|
| | Ipsi | Contra | Ipsi | Contra | Ipsi | Contra | Ipsi | Contra |
| Preoperational (n = 15) | 109 ± 3 | 97 ± 2 | 106 ± 2 | 97± 3 | 48 ± 6 | 61 ± 5 | 51 ± 7 | 59± 7 |
| DBS on (n = 15) | 109 ± 3 | 93 ± 2 | 107 ± 2 | 95 ± 2 | 41 ± 5 | 66 ± 5 | 48 ± 6 | 58± 8 |
| DBS off (n = 12) | 112 ± 6 | 93 ± 3 | 105 ± 4 | 98 ± 4 | 39 ± 6 | 58 ± 6 | 36 ± 4 | 42± 8 |

contralateral to the stimulated ear following the operation, resulting in increased ipsilateral-contralateral latency difference after correction of the hearing loss [29]. Normalization of the intrahemispheric asynchrony after behavioral compensation with sound amplification, demonstrated by increased ipsilateral-contralateral latency difference, was recently observed in patients suffering from unilateral hearing loss [30], qualitatively similarly to our current results in PD patients. The observed changes were attributed to plasticity of the auditory system for adapting to the changed auditory environment.

Earlier AEF studies have suggested a direct auditory cortical disruption by PD (*e.g.*, [26]), possibly related to basal ganglia dysfunction together with the emphasized sensorineural hearing loss in PD [25]. Furthermore, local field potentials recorded from the STN are correlated with spontaneous ~10-Hz oscillatory activity over the auditory temporal cortices [45, 46]. PD patients have a clear defect in psychophysical detection of very short temporal gaps within noise bursts, suggested to be related to impaired detection of amplitude modulations in the auditory cortex [47]. This deficit has been shown, to some extent, to be compensated with DBS but not with levodopa therapy, suggesting that it is not related to the dopaminergic deficit in PD as such [47]. Moreover, the implantation of STN DBS significantly improved, both with DBS on and off, the abnormal stimulus frequency-related gating of P1/N1 auditory evoked potentials (AEPs) of PD patients observed before the operation [48]. The observed effect of DBS was attributed to top-down modulation from the frontal cortex on the temporal auditory areas [48]. Again, levodopa dosage had no effect on the AEPs.

In our patients after the 6-month DBS therapy, the ipsi-contralateral differences of N100m response latencies during DBS on were larger than in the preoperative baseline measurements. This suggests that DBS induced here analogous plastic changes in the auditory system to the correction of unilateral hearing loss [29, 30]. Our finding supports the notion that DBS can induce gradual reorganization of neural circuits through enhanced synaptic plasticity and neurogenesis [11, 12]. Direct anatomical projections between auditory cortex and STN are sparse or absent in animal models [49, 50], but output pathways from the caudal pallidum to auditory pathways, *e.g.*, to the inferior colliculus, the medial geniculate nucleus, and the temporal cortex have been reported [51]. In humans, the basal ganglia "gate" auditory inputs at various levels [52]. The effects of STN DBS on AEFs are probably mediated through such an indirect route.

The study has some limitations to be considered when interpreting the findings. In all patients, the MEG measurements were done first with DBS on and then DBS off. Shifting the patient from under the dewar, turning the stimulator off, repositioning the patient and re-localization of the head position took approximately 10 minutes. DBS off time was thus relatively short: At least three hours of STN DBS off is usually considered to be required to establish a steady motor DBS off state for efficacy studies [53], and ~50% of the total change in the motor scales has been estimated to occur within 5 min after DBS is turned off [54]. We, however, decided to exclude any comparisons between the DBS on and off conditions in the present study. Minor changes in the sound intensities between measurement conditions are

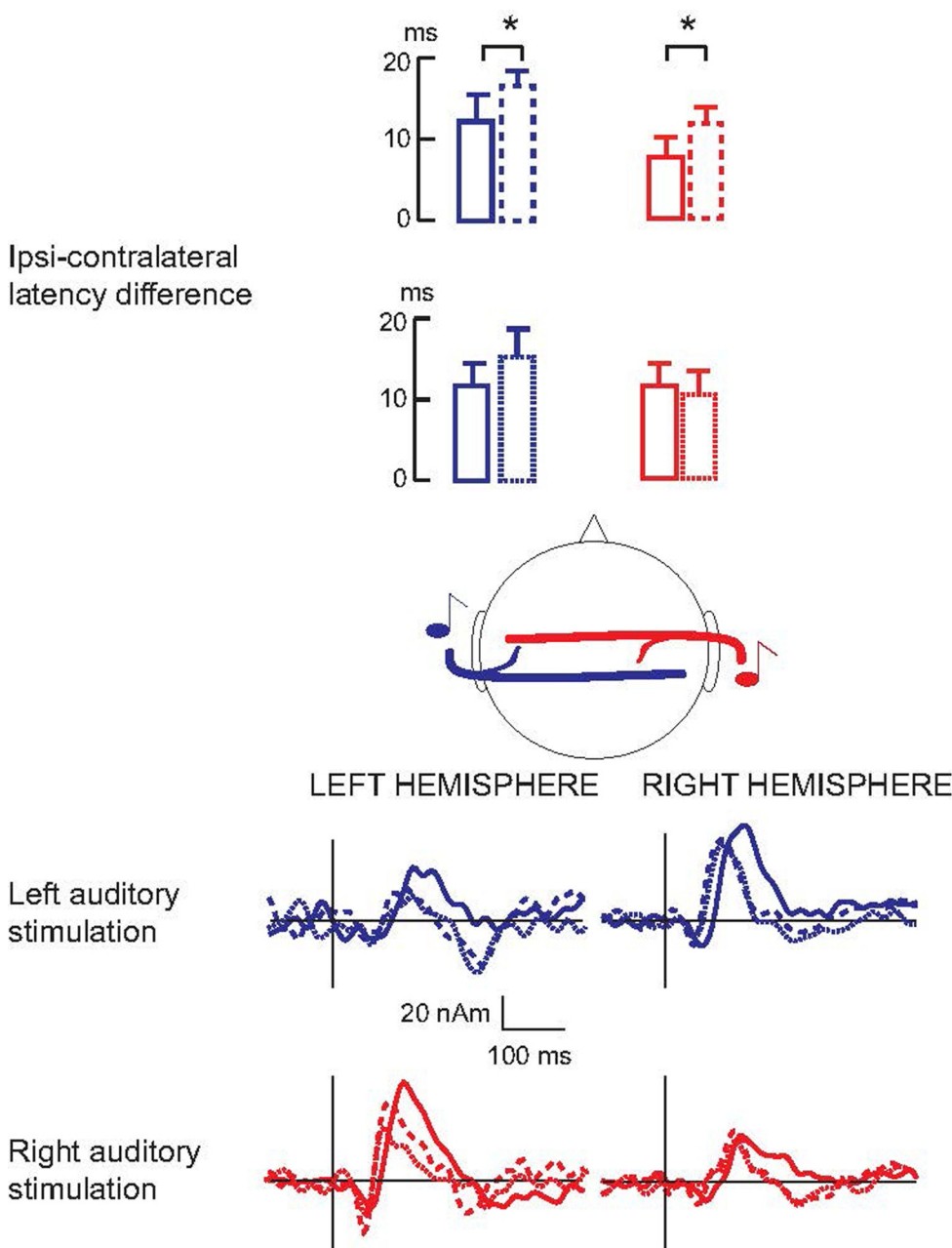

**Fig 2. The interhemispheric latency difference increased from preoperative to postoperative DBS on condition.**
Bottom: N100m source strengths as a function of time in one subject to both left- (blue) and right-sided (red) auditory
stimulation in preoperative (full line), DBS on (dashed line), and DBS off (dotted line) conditions. Top: Comparison of
the interhemispheric latency differences in both hemispheres in preoperative (full line) and DBS on (dashed line)
conditions (n = 15), and in preoperative (full line) and DBS off (dotted line) conditions for the subjects who tolerated
DBS off condition (n = 12).

possible, but very unlikely to affect our results on the N100m latencies that are known to satu-
rate at sound intensities above 50 dB HL [55]. Although the number of patients in our study is
in the range of previous reports of the effects of DBS on brain electrophysiology, future studies
on larger patient populations and in different sensory systems are needed to better understand
the neuronal reorganization related to DBS in PD.

## Conclusions

Our results demonstrate that MEG can be used to follow possible modulations of cortical evoked activity related to DBS in PD patients. Particularly, the present results suggest that the DBS normalizes neuronal asynchrony in the central auditory pathways, reflected here as increased contra-ipsilateral N100m response latency differences compared with the preoperative state. MEG can thus provide important insight into DBS-induced plastic changes and reorganization of non-motor neural networks.

## Author Contributions

**Conceptualization:** Kati Valkonen, Jyrki P. Mäkelä, Katja Airaksinen, Riku Kivisaari, Eero Pekkonen.

**Formal analysis:** Kati Valkonen, Jussi Nurminen, Hanna Renvall.

**Funding acquisition:** Eero Pekkonen.

**Investigation:** Kati Valkonen, Jyrki P. Mäkelä, Katja Airaksinen.

**Methodology:** Jyrki P. Mäkelä, Jussi Nurminen, Riku Kivisaari, Hanna Renvall.

**Project administration:** Jyrki P. Mäkelä, Eero Pekkonen.

**Resources:** Katja Airaksinen, Eero Pekkonen.

**Software:** Jussi Nurminen.

**Supervision:** Jyrki P. Mäkelä, Katja Airaksinen, Hanna Renvall, Eero Pekkonen.

**Visualization:** Kati Valkonen, Jussi Nurminen, Hanna Renvall.

**Writing – original draft:** Kati Valkonen.

**Writing – review & editing:** Kati Valkonen, Jyrki P. Mäkelä, Katja Airaksinen, Jussi Nurminen, Riku Kivisaari, Hanna Renvall, Eero Pekkonen.

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
