## [Decision Letter · Decision Letter 0]

21 Oct 2021

PONE-D-21-16394

Deep brain stimulation of subthalamic nucleus modulates cortical auditory processing in advanced Parkinson’s Disease

PLOS ONE

Dear Dr. Renvall,

Thank you for submitting your manuscript to PLOS ONE. After careful consideration, we feel that it has merit but does not fully meet PLOS ONE’s publication criteria as it currently stands. Therefore, we invite you to submit a revised version of the manuscript that addresses the points raised during the review process.

We look forward to receiving your revised manuscript.

Kind regards,

Tuhin Virmani

Academic Editor

PLOS ONE

Journal Requirements:

"KV: Finnish Parkinson Foundation

KV: University of Helsinki

HR: grant 321460, Academy of Finland 

We note that one or more of the authors is affiliated with the funding organization, indicating the funder may have had some role in the design, data collection, analysis or preparation of your manuscript for publication; in other words, the funder played an indirect role through the participation of the co-authors. If the funding organization did not play a role in the study design, data collection and analysis, decision to publish, or preparation of the manuscript and only provided financial support in the form of authors' salaries and/or research materials, please do the following:

a. Review your statements relating to the author contributions, and ensure you have specifically and accurately indicated the role(s) that these authors had in your study. These amendments should be made in the online form.

b. Confirm in your cover letter that you agree with the following statement, and we will change the online submission form on your behalf: 

“The funder provided support in the form of salaries for authors [insert relevant initials], but did not have any additional role in the study design, data collection and analysis, decision to publish, or preparation of the manuscript. The specific roles of these authors are articulated in the ‘author contributions’ section.

"This study was supported by a grant of University of Helsinki (to KV), Finnish Parkinson Foundation (KV), and Academy of Finland (grant number 321460, HR)."

"This study was supported by a grant of University of Helsinki (to KV), Finnish Parkinson Foundation (KV), and Academy of Finland (grant number 321460, HR)."

Reviewers' comments:

Reviewer's Responses to Questions

**Comments to the Author**

1. Is the manuscript technically sound, and do the data support the conclusions?

Reviewer #1: Yes

2. Has the statistical analysis been performed appropriately and rigorously? 

Reviewer #1: Yes

3. Have the authors made all data underlying the findings in their manuscript fully available?

Reviewer #1: No

4. Is the manuscript presented in an intelligible fashion and written in standard English?

Reviewer #1: Yes

5. Review Comments to the Author

Reviewer #1: SUMMARY:

This study investigated auditory cortical processing in PD patients before and after receiving deep brain stimulation (DBS). Short tone pips (50 ms, 1 kHz) were administered to each ear independently, and equivalent current dipoles were estimated for the auditory evoked fields (AEFs) around 100 ms based on a subset of channels in each hemisphere. UPDRS-III and LEDD were also measured both pre-op and post-op (UPDRS-III and MEG was measured for both DBS-ON and -OFF post-op). The latency difference between contra- and ipsilateral AEF peaks was found to differ between pre-op and DBS-ON post-op. The authors interpret this difference as increased neuronal synchrony in the auditory pathways mediated by DBS.

GENERAL COMMENTS:

The dataset containing DBS-pre-op and -post-op MEG recordings for PD patients should be commended in and by itself since disease heterogeneity is a substantial challenge in DBS research. The manuscript is clearly written, however it lacks some coherence in the Introduction, in particular. The topic of DBS effects on auditory cortical processing is certainly relevant in the context of both general PD research as well as research on neuromodulation (and DBS in particular).

My main concern lies with the choice of dependent variables to analyze (and not to analyze) which I have explained in more detail in the MAJOR CONCERNS section below.

PRACTICAL COMMENTS:

Line numbers would have been helpful.

MAJOR CONCERNS:

The latency analyses are not motivated in the Introduction. I suggest dedicating substantial space In the Intro to motivate this analysis since I suspect this was not merely a post-hoc justified analysis.

Instead, amplitude differences are briefly discussed in the Introduction. However, there are no amplitude analyses performed in the manuscript. I therefore suggest also performing amplitude analyses.

MINOR COMMENTS:

ABSTRACT:

"Magnetoencephalography provides a feasible tool to study DBS-induced plastic changes and reorganization of both motor and non-motor neural networks in PD" -> I don't see exactly how this is shown or evaluated in this paper, however, I would strongly recommend the authors to include evidence and visualization(s) that could allow for such a conclusion

INTRO:

In general, the Introduction lacks a bit of coherence - there's almost no logical transition from paragraph to paragraph: 'Degeneration of dopaminergic neurons' > 'DBS is an effective treatment' (no other treatments mentioned or at what stage DBS is usually administered as treatment); 'No neural plasticity induced by DBS' > 'DBS causes strong artifacts in MEG' (why is MEG relevant in this context?); 'measuring cortical modulations generated by DBS' > 'PD patients suffering from impaired hearing' -> besides the already mentioned points regarding stronger motivation for the latency analyses, it seems relevant to also briefly introduce DBS effects on cortical processing and non-motor symptoms in PD more broadly in order to better place the current study in that context.

"The PD patients often suffer from a broad spectrum of non-motor signs" -> "PD patients often suffer from a broad spectrum of non-motor signs"

"most prominent auditory evoked fields (AEF) at ~100 ms" -> "most prominent auditory evoked fields (AEF[s]) at ~100 ms"

"produce well-characterized auditory evoked fields [23-24]." -> "produce well-characterized [AEF]s [23-24]." (Abbrev is introduced two sentences earlier)

MATERIALS AND METHODS:

"gave an informed written consent" -> "gave informed written consent"

"Three of the remaining 15 patients did not tolerate MEG measurement at six months when medication was off" -> I'm assuming the authors here mean "when DBS was off" and not "medication"?

"The baseline MEG was measured before the DBS implantation, and the follow-up measurements were conducted at about seven months (see above) after the operation, both with DBS on and off." -> based on hints elsewhere in the text, I'm assuming all three recordings were with medication on? This should be clarified in the text

"The DBS frequency was adjusted to 130 Hz before MEG measurements to avoid interference with the head position indicator (HPI) coil signals" -> were all patients kept on their original bi-/monopolar stimulation settings, and if so, did differences in bi-/monopolar settings not affect SNR in the individual patients (after tSSS)?

"twice in a row, to exclude formation" -> "twice in a row to exclude formation"

"About 100 artifact-free auditory responses were averaged per stimulated ear." -> please provide mean and SD for number of trials across participants

"The strongest 100-ms AEFs (N100m) we first scrutinized with a single sensor analysis." -> word order/syntax is slightly off.

"The peak response amplitudes were determined by finding the absolute maxima of evoked signals in a time window 80 - 130 ms after the stimulus onset." -> in gradiometers and/or magnetometers?

"the vector sum √((/)^2 + (/)^2) of the channel pair showing the maximum signal." -> even tho vector addition is relatively straightforward, I suggest still specifying what the terms refer to in terms of the gradiometer signals in question when the authors decide to include the actual equation

"In signal strength comparisons, the vector sums simplify the analysis when the orientation of the neural current changes as a function of time, with only minor accompanying changes in the source location. In such a case, the amplitude measurements from any single channel can be misleading." -> I'm assuming this point is made in relation to combining info from gradiometer pairs (in the (Elekta) Vectorview system), but for those less familiar with the orthogonally oriented gradiometer coils, perhaps integrate a few direct references to the make-up of the sensors in question to help less familiar readers a better grasp of the procedure?

"Subsequently, the cortical sources of the N100m responses were searched from contra- and ipsilateral hemispheres using a subset of 10-15 gradiometer pairs around the locus of the maximum response by means of guided current modeling (equivalent current dipole [ECD]; [28]), separately for each subject." -> was the maximum response(s) determined in each hemisphere separately? Please clarify and adjust description earlier accordingly. Also, what was the source of variation between 10-15 gradiometer pairs in the subsets?

"Applying the same source model in each data set minimizes variation due to possible differences between source models;" -> I'm having a bit of difficulty reconciling this with the fact that the ECDs were estimated for pre-DBS and post-DBS separately - perhaps the ECD selection procedure is the mediating link here, but then I'm a bit puzzled that the SNR is used as a selection criterion since I would imagine that this would favor the pre-DBS ECD location... could the authors clarify this procedure a bit?

"Statistical comparisons of the latency differences between the preoperative AEFs and those obtained with DBS on and DBS off were performed using non-parametric sign test" -> could the authors elaborate on why? This seems a bit peculiar to me given that these are latency values...

And how were the amplitudes statistically evaluated - I would assume by repeated measures ANOVA, but not clear from the Results either?

"Bonferroni corrected for number of comparisons." -> how many comparisons were corrected for?

"The results are reported as mean ± standard error of mean." -> this doesn't align very well with the implementation of non-parametric sign tests.

RESULTS:

"After artifact removal by tSSS, sources of AEFs were analyzable in both hemispheres in all 15 patients" -> plz provide some means of visualizing the effect of tSSS, as well as elaborate a bit on what is meant by "analyzable" and how this was evaluated. Furthermore, was the effects of tSSS validated by any means, i.e. in order to ensure that MEG signals remaining after tSSS were not still to some extent contaminated by the DBS artefacts?

"At the source level, the N100m responses peaked at 97 ± 3 ms and 109 ± 3 ms in the left, and at 106 ± 2 ms and 97 ± 2 ms in the right auditory cortex for the right- and left-sided stimulation in the preoperational condition, and at 95 ± 2 ms and 109 ± 3 ms in the left, and at 107 ± 2 ms and 93 ± 2 ms in the right auditory cortex for the right- and left-sided stimulation in the “DBS on” condition. The ipsi-contralateral difference of N100m peak latencies significantly increased from the preoperative to “DBS on” condition (pooled across the stimulated ears, 10 ± 2 ms vs. 14 ± 1 ms: p = 0.036; see Fig 2)." -> This info would work really well in a table (also including the DBS off condition); surprisingly hard to follow in text form. Also, the text presentation would benefit from some reorganization to better reflect the logic of the analyses/comparisons performed (i.e. I suggest grouping the contra- and ipsi-lateral values together since those are the values that are directly compared).

The signficant effects should be tested post-hoc regarding whether contra-latency decreased or ipsi-latency increased (the implicit results of these are referred to in the Discussion and Abstract, but not tested for or explicitly stated in the Results section).

"In the “DBS off” condition (data available from 12 subjects), the N100m responses peaked at 98 ± 4 ms and112±6ms in the left, and at 105±4ms and 93±3 ms in the right auditory cortex for the right- and left-sided stimulation." -> And what were the latency values for the PRE condition for those 12 subjects? Also, were DBS-ON and -OFF not compared?

"daily dose (LEDD) appeared to decrease" -> "daily dose (LEDD) appeared to decrease[d]"

DISCUSSION:

"been shown, to some extent, be compensated with DBS" -> "been shown, to some extent, [to] be compensated with DBS"

"plastic changes in the auditory system than the correction of unilateral hearing loss [25]." -> "plastic changes in the auditory system [to] the correction of unilateral hearing loss [25]."

"Specifically, the decreased contralateral N100m latency here suggests that the effect of DBS is reflected primarily as increased neuronal synchrony (cf. [40])." -> This is not shown in the Results - see my comment for the relevant part of the Results section

FIGURES:

FIG 1:

"The arrow indicates the stimulated (left) ear." -> I don't see any arrows in the figure...

-> I suggest clarifying which of the highlighted sensors are gradiometers and which are magnetometer(s)

-> I also suggest adding "ipsilateral" and "contralateral" (or similar) labels above the two channel highlights

-> if at all possible, would be lovely to see the traces in the highlighted sensors including the variance (e.g. via shading or similar)

FIG 2:

"The interhemispheric latency difference increased from preoperative to postoperative DBS on condition." -> "The interhemispheric latency difference increased from preoperative to postoperative DBS on condition."

-> Legend missing - not immediately clear what full, dotted and broken lines refer to

-> Top: are the values for n=15 for PRE and DBS-ON and then for n=12 for DBS-OFF, cuz then I suggest splitting up into two different plots - one for PRE and DBS-ON for n=15, and one for all three conditions for n=12 (or similar) - but not for different n's in the same plot.

-> Bottom: would be very nice to see these traces for the group average (incl. variance estimates) and not only for a single subject.

6. PLOS authors have the option to publish the peer review history of their article (what does this mean?). If published, this will include your full peer review and any attached files.

Reviewer #1: **Yes: **Andreas Højlund

---

## [Author Response · Author response to Decision Letter 0]

15 Dec 2021

Replies to the Reviewer's comments

We thank the reviewer for the very useful comments that have helped us to greatly improve our manuscript. Please find below our detailed answers to the reviewer’s questions and comments. We hope that our manuscript can now be accepted for publication in PLOS One.

The latency analyses are not motivated in the Introduction. I suggest dedicating substantial space In the Intro to motivate this analysis since I suspect this was not merely a post-hoc justified analysis.

We indeed hypothesized that especially the auditory response latencies would be affected, taken the existing data on the reorganization of central auditory pathways after hearing loss (Vasama et al. 1998, Chang et al. 2021). Furthermore, the amplitudes of auditory evoked responses are known to inherently display considerable interindividual variability (Mäkelä et al. 1993, Renvall et al. 2012) and be sensitive to head movements in successive measurements (Nenonen et al. 2010). We have now better motivated the choice of analysis (see p. 4). 

Instead, amplitude differences are briefly discussed in the Introduction. However, there are no amplitude analyses performed in the manuscript. I therefore suggest also performing amplitude analyses.

We have now added the results of the amplitude analyses to Table 2 in Results section (p. 10).

MINOR COMMENTS:

ABSTRACT:

"Magnetoencephalography provides a feasible tool to study DBS-induced plastic changes and reorganization of both motor and non-motor neural networks in PD" -> I don't see exactly how this is shown or evaluated in this paper, however, I would strongly recommend the authors to include evidence and visualization(s) that could allow for such a conclusion

We have modified this statement to “Whole-head magnetoencephalography provides a feasible tool to study DBS-induced motor and non-motor neural networks in PD, and to track possible changes related to cortical reorganization or plasticity induced by DBS”.

INTRO:

In general, the Introduction lacks a bit of coherence… 

 'Degeneration of dopaminergic neurons' > 'DBS is an effective treatment' (no other treatments mentioned or at what stage DBS is usually administered as treatment); 

'No neural plasticity induced by DBS' > 'DBS causes strong artifacts in MEG' (why is MEG relevant in this context?); 

'measuring cortical modulations generated by DBS' > 'PD patients suffering from impaired hearing' -> 

We have now modified the text throughout Introduction, for improving coherence between the paragraphs.

… it seems relevant to also briefly introduce DBS effects on cortical processing and non-motor symptoms in PD more broadly in order to better place the current study in that context.

We have added references to Georgiev et al. 2021 on DBS effects to sleep and anxiety (p. 3), and to Hyder et al. 2021 on DBS effects to speech sounds (p. 4).

"The PD patients often suffer from a broad spectrum of non-motor signs" -> "PD patients often suffer from a broad spectrum of non-motor signs"

"most prominent auditory evoked fields (AEF) at ~100 ms" -> "most prominent auditory evoked fields (AEF[s]) at ~100 ms"

"produce well-characterized auditory evoked fields [23-24]." -> "produce well-characterized [AEF]s [23-24]." (Abbrev is introduced two sentences earlier)

All corrected.

MATERIALS AND METHODS:

"gave an informed written consent" -> "gave informed written consent"

Corrected.

"Three of the remaining 15 patients did not tolerate MEG measurement at six months when medication was off" -> I'm assuming the authors here mean "when DBS was off" and not "medication"? 

Corrected to “DBS were off”. 

"The baseline MEG was measured before the DBS implantation, and the follow-up measurements were conducted at about seven months (see above) after the operation, both with DBS on and off." -> based on hints elsewhere in the text, I'm assuming all three recordings were with medication on? This should be clarified in the text

Medication was indeed on, and this has now been clarified in the text (p. 6).

"The DBS frequency was adjusted to 130 Hz before MEG measurements to avoid interference with the head position indicator (HPI) coil signals" -> were all patients kept on their original bi-/monopolar stimulation settings, and if so, did differences in bi-/monopolar settings not affect SNR in the individual patients (after tSSS)?

All patients were kept at their original stimulation settings. Monopolar DBS induces more high-frequency artifacts than bipolar one, but they can be effectively removed by filtering (see, e.g., Airaksinen et al. 2011). This information is now added to the Methods (p. 7).

"twice in a row, to exclude formation" -> "twice in a row to exclude formation"

Corrected.

"About 100 artifact-free auditory responses were averaged per stimulated ear." -> please provide mean and SD for number of trials across participants

Added (p. 7).

"The strongest 100-ms AEFs (N100m) we first scrutinized with a single sensor analysis." -> word order/syntax is slightly off.

The sentence has been changed to “The 100-ms AEFs (N100m) we first analyzed at the sensor level”.

"The peak response amplitudes were determined by finding the absolute maxima of evoked signals in a time window 80 - 130 ms after the stimulus onset." -> in gradiometers and/or magnetometers?

“At the gradiometer channels” added to the text (p. 8).

"the vector sum √((/)^2 + (/)^2) of the channel pair showing the maximum signal." -> even though vector addition is relatively straightforward, I suggest still specifying what the terms refer to in terms of the gradiometer signals in question when the authors decide to include the actual equation

Word “gradiometer” added to clarify the equation (p. 8).

"In signal strength comparisons, the vector sums simplify the analysis when the orientation of the neural current changes as a function of time, with only minor accompanying changes in the source location. In such a case, the amplitude measurements from any single channel can be misleading." -> I'm assuming this point is made in relation to combining info from gradiometer pairs (in the (Elekta) Vectorview system), but for those less familiar with the orthogonally oriented gradiometer coils, perhaps integrate a few direct references to the make-up of the sensors in question to help less familiar readers a better grasp of the procedure?

References to Hämäläinen et al. 1993 and Renvall and Hari (2002, with similar approach) added.

"Subsequently, the cortical sources of the N100m responses were searched from contra- and ipsilateral hemispheres using a subset of 10-15 gradiometer pairs around the locus of the maximum response by means of guided current modeling (equivalent current dipole [ECD]; [28]), separately for each subject." -> was the maximum response(s) determined in each hemisphere separately?

Yes, it was. This information is now added to the text (p. 8).

Also, what was the source of variation between 10-15 gradiometer pairs in the subsets?

10-15 channel pairs were selected to adequately cover the area of maximum source; this part of the text has now been clarified (p. 8).

"Applying the same source model in each data set minimizes variation due to possible differences between source models;" -> I'm having a bit of difficulty reconciling this with the fact that the ECDs were estimated for pre-DBS and post-DBS separately - perhaps the ECD selection procedure is the mediating link here, but then I'm a bit puzzled that the SNR is used as a selection criterion since I would imagine that this would favor the pre-DBS ECD location... could the authors clarify this procedure a bit?

We apologize for the previously unclear paragraph. We have now reorganized it and added text to clarify the procedure. The ECDs were searched in all three conditions (preoperational, DBS on, DBS off) separately, and of those ECDs the one with best SNR was chosen to represent the source in all conditions (p. 8-9). 

"Statistical comparisons of the latency differences between the preoperative AEFs and those obtained with DBS on and DBS off were performed using non-parametric sign test" -> could the authors elaborate on why? This seems a bit peculiar to me given that these are latency values…

"Bonferroni corrected for number of comparisons." -> how many comparisons were corrected for?

We wanted to use a statistical test which is free of assumptions for distribution, taken that AEF latency values typically show little interindividual variability and we were especially interested in the latency differences between the hemispheres, and thus an assumption of normal distribution would have been strong. The numbers were Bonferroni corrected by 2 (preoperational vs. DBS on, and preoperational vs. DBS off). These parts have been clarified in the text (p. 9). 

And how were the amplitudes statistically evaluated - I would assume by repeated measures ANOVA, but not clear from the Results either?

The amplitude values were not statistically evaluated (see the response above on their large interindividual variability and sensitivity to head movements).

"The results are reported as mean ± standard error of mean." -> this doesn't align very well with the implementation of non-parametric sign tests.

The statistical testing was not done on the absolute latency and amplitude values, but on the interhemispheric latency differences (see above). After consideration we have decided to keep the mean ±SEM values in the Table, as we think that they may tell more to the reader about the variability between subjects than, e.g., median value and mean absolute deviation would do. 

RESULTS:

"After artifact removal by tSSS, sources of AEFs were analyzable in both hemispheres in all 15 patients" -> plz provide some means of visualizing the effect of tSSS, as well as elaborate a bit on what is meant by "analyzable" and how this was evaluated. Furthermore, was the effects of tSSS validated by any means, i.e. in order to ensure that MEG signals remaining after tSSS were not still to some extent contaminated by the DBS artefacts?

The effect of tSSS on AEFs has been visualized and discussed in Airaksinen et al. (2011), and we now refer the reader to this publication (p. 7).

"At the source level, the N100m responses peaked at 97 ± 3 ms … -> This info would work really well in a table (also including the DBS off condition); surprisingly hard to follow in text form. Also, the text presentation would benefit from some reorganization to better reflect the logic of the analyses/comparisons performed (i.e. I suggest grouping the contra- and ipsi-lateral values together since those are the values that are directly compared).

We thank the reviewer for this suggestion which definitely clarifies the Results section. The latency values have now been transferred to Table 2, and the text has been reorganized.

The significant effects should be tested post-hoc regarding whether contra-latency decreased or ipsi-latency increased (the implicit results of these are referred to in the Discussion and Abstract, but not tested for or explicitly stated in the Results section).

In the light of the most recent results (Chang et al. 2021), we consider our results to better relate to normalization of asynchrony in the auditory pathways with DBS than to specific changes in only contra- or ipsilateral auditory tracks. We have now modified the corresponding text in Abstract and Discussion, and excluded any direct references to contralateral latency values. 

"In the “DBS off” condition (data available from 12 subjects), the N100m responses peaked at 98 ± 4 ms and112±6ms in the left, and at 105±4ms and 93±3 ms in the right auditory cortex for the right- and left-sided stimulation." -> And what were the latency values for the PRE condition for those 12 subjects? Also, were DBS-ON and -OFF not compared?

We decided to exclude any comparisons between the DBS on and off conditions in the present study, taken that the DBS-off time before the MEG measurement was short. This is discussed at p. 13.

"daily dose (LEDD) appeared to decrease" -> "daily dose (LEDD) appeared to decrease[d]"

Corrected.

DISCUSSION:

"been shown, to some extent, be compensated with DBS" -> "been shown, to some extent, [to] be compensated with DBS"

"plastic changes in the auditory system than the correction of unilateral hearing loss [25]." -> "plastic changes in the auditory system [to] the correction of unilateral hearing loss [25]."

Corrected.

"Specifically, the decreased contralateral N100m latency here suggests that the effect of DBS is reflected primarily as increased neuronal synchrony (cf. [40])." -> This is not shown in the Results - see my comment for the relevant part of the Results section

See above: we have now excluded any direct references to contralateral latency values and changed the wording to “normalization of neuronal asynchrony” (Abstract and Conclusions, p. 13).

FIGURES:

FIG 1:

"The arrow indicates the stimulated (left) ear." -> I don't see any arrows in the figure...

-> I suggest clarifying which of the highlighted sensors are gradiometers and which are magnetometer(s)

-> I also suggest adding "ipsilateral" and "contralateral" (or similar) labels above the two channel highlights

We have clarified the Figure and added information on sensor types to the Figure legend.

-> if at all possible, would be lovely to see the traces in the highlighted sensors including the variance (e.g. via shading or similar)

We have now included an insert demonstrating the single-trace responses on one (maximum) gradiometer channel in preoperational condition.

FIG 2:

-> Legend missing - not immediately clear what full, dotted and broken lines refer to

Corrected.

-> Top: are the values for n=15 for PRE and DBS-ON and then for n=12 for DBS-OFF, I suggest splitting up into two different plots - one for PRE and DBS-ON for n=15, and one for all three conditions for n=12 (or similar) - but not for different n's in the same plot.

Corrected.

-> Bottom: would be very nice to see these traces for the group average (incl. variance estimates) and not only for a single subject.

We considered this suggestion, but due to the first author’s current position elsewhere, the producing such a figure became difficult in practice. If the Reviewer still thinks the Figure to benefit from this addition, we will produce it.

---

## [Decision Letter · Decision Letter 1]

9 Feb 2022

Deep brain stimulation of subthalamic nucleus modulates cortical auditory processing in advanced Parkinson’s Disease

PONE-D-21-16394R1

Dear Dr. Renvall,

We’re pleased to inform you that your manuscript has been judged scientifically suitable for publication and will be formally accepted for publication once it meets all outstanding technical requirements.

Kind regards,

Tuhin Virmani

Academic Editor

PLOS ONE

Additional Editor Comments (optional):

Reviewers' comments:

Reviewer's Responses to Questions

**Comments to the Author**

1. If the authors have adequately addressed your comments raised in a previous round of review and you feel that this manuscript is now acceptable for publication, you may indicate that here to bypass the “Comments to the Author” section, enter your conflict of interest statement in the “Confidential to Editor” section, and submit your "Accept" recommendation.

Reviewer #1: All comments have been addressed

2. Is the manuscript technically sound, and do the data support the conclusions?

Reviewer #1: Yes

3. Has the statistical analysis been performed appropriately and rigorously? 

Reviewer #1: Yes

4. Have the authors made all data underlying the findings in their manuscript fully available?

Reviewer #1: No

5. Is the manuscript presented in an intelligible fashion and written in standard English?

Reviewer #1: Yes

6. Review Comments to the Author

Reviewer #1: The authors have fully addressed all my concerns, and I find the manuscript to have greatly improved.

7. PLOS authors have the option to publish the peer review history of their article (what does this mean?). If published, this will include your full peer review and any attached files.

Reviewer #1: **Yes: **Andreas Højlund

---

## [Editor Report · Acceptance letter]

14 Feb 2022

PONE-D-21-16394R1 

Deep brain stimulation of subthalamic nucleus modulates cortical auditory processing in advanced Parkinson’s Disease 

Dear Dr. Renvall:

I'm pleased to inform you that your manuscript has been deemed suitable for publication in PLOS ONE. Congratulations! Your manuscript is now with our production department. 

Kind regards, 

on behalf of

Dr. Tuhin Virmani 

Academic Editor

PLOS ONE